# Suggestive Labelling for Medical Image Analysis by Adaptive Latent Space Sampling

## Abstract

Supervised deep learning for medical imaging analysis requires a large amount of training samples with annotations, which are expensive and time-consuming to obtain. During the training of a deep neural network, the annotated samples are fed into the network in mini-batches and often regarded of equal importance. However, some samples may become less informative during training as the gradients of the loss function vanish. Other samples of higher utility or hardness may be more demanded for training and require more exploitation. We propose a novel training framework which adaptively selects informative samples that are fed to the training process. The adaptive selection or sampling is performed based on a hardness-aware strategy in the latent space constructed by a generative model. To evaluate the proposed idea, we perform experiments on a medical image dataset IVUS for pixel-wise regression task. We demonstrate that the proposed method can reduce the number of required training samples and save effort for annotation process.

**Keywords:** Data Efficiency, Deep Learning.

## 1. Introduction

Recent advances in deep learning have been successful in delivering the state-of-the-art (SOTA) performance in a variety of areas including computer vision, nature language processing, etc. Not only do advanced network architecture designs and better optimization techniques contribute to the success, but the availability of large annotated datasets (e.g. ImageNet (Deng et al., 2009), MS COCO (Lin et al., 2014), Cityscapes (Cordts et al., 2016)) also plays an important role. However, collecting unlabeled data and the subsequent annotating process are both expensive and time-consuming. For example, it takes hours for an experienced radiologist to segment the brain tumor on medical images for just one case. In particular, for medical imaging applications, the annotation is often limited by the available resources of expert analysts and data protection issues, which makes it even more challenging for curating large datasets. In this paper, we propose a model state-aware framework for data-efficient deep representation learning. The main idea is to mine 'harder' training samples progressively on the data manifold according to the current parameter state of a network until a certain criteria is fulfilled.

## 2. Methodology

The proposed training framework consists of two stages as shown in Figure 1. The first stage can be considered as a preparation phase where a VAE-based generative model [1] is trained

---

1. Here we use VAE as an example to demonstrate the idea. Other kinds of generative models can be also used in the framework accordingly.

using unannotated samples. We obtain the generator (decoder) $G$ and the encoder $E$ with trained and fixed parameters as well as the $n$-dimensional latent space $\mathbb{R}^n$. In the second

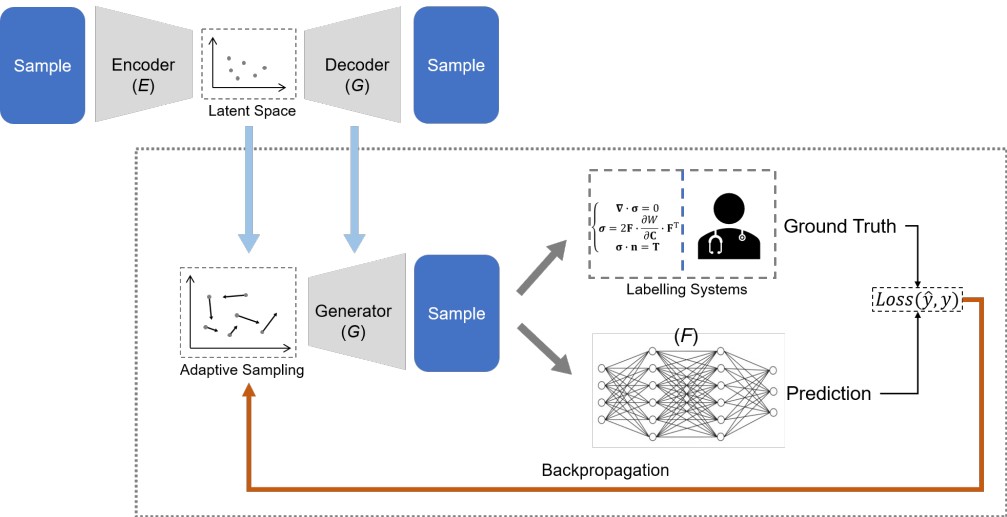

Figure 1: The general pipeline of proposed framework. Stage 1, preparation, is located at the top left corner. Stage 2, training, is located within the dashed rectangle.

stage, 'hard' samples are mined iteratively in the latent space and fed into the network for training. The sampling strategies are based on the error information (normalized gradients) of the neural network model $F$ back-propagated through the generator $G$. More specifically, the proposed method iteratively selects those relatively more informative training samples from the latent space and add them into an incremental training set. In the first round, a set of random points in the latent space are used for generating training samples which are annotated by the labeling tool. The labeling tool can be either a human expert or intelligent system according to the task [2]. The neural network model $F$ is first trained for a couple of epochs with the current training set using a given loss function. Then, we randomly select a few annotated samples from the training set and identify the embedding positions in latent spaces. We evaluate the given loss function with those samples and calculate the gradients for each embedding point in the latent space using back-propagation. Once the gradients are derived for each embedding point in latent space, we can move the embedding points along with the directions of gradients using a pre-defined step size such that the generator $G$ can produce more informative samples using the new position and adding them into the existing training set for the next training epoch.

## 3. Experimental Results

The main purpose of experiments is to demonstrate the effectiveness of our proposed framework which can achieve the same performance using less training data. Thus, the perfor-

---

2. In our study, the labelling system is a finite element analysis solver (FEA solver)

mance of a given target model was evaluated using the same experimental setting (i.e. same optimizer, number of training iteration, batch size and size of training data). The baseline method employed in this paper randomly selects a subset of training samples from real data until a certain condition is fulfilled, namely a predefined size of training samples.

Intravascular ultrasound (IVUS) is a catheter-based medical imaging modality to identify the morphology and composition of atherosclerotic plaques of the blood vessels. Imaged-based biophysical simulation of coronary plaques is used for assessing the structural stress within the vessel wall (Figure 2(a)), which relies on time-consuming finite element analysis (FEA) (Teng et al., 2014). Here, we aim to train a deep neural network to approximate the FEA, which takes medical image as input and predicts a structural stress map. Our IVUS dataset consist of 1,200 slices of 2D gray-scale images of coronary plaques and corresponding vessel wall segmentation from the vendor. An in-house Python package serves as a labeling tool by solving partial differential equations (PDEs) on segmentation masks.

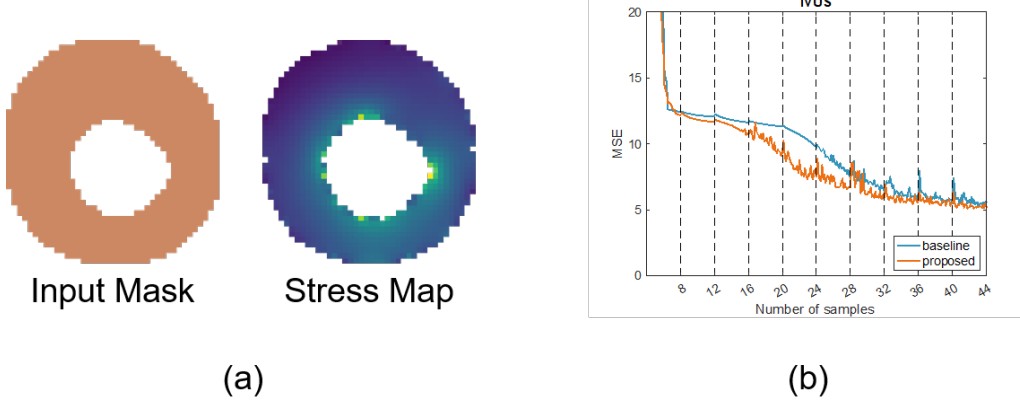

Figure 2: (a)An example of the input and output of the labeling tool for IVUS dataset. (b) Comparison of the baseline method and the proposed method using different number of training samples

For the experiment, we progressively increased the size of train set and reported the accuracy and mean square error (MSE) on the independent test sets. Experiment was repeated for five times for plotting the mean and variance (Figure 2(b)). It is observed that the target model trained under the proposed framework yields a better performance than the baseline method until the target model reaches the bottleneck where the performance can hardly be improved by increasing size of the training samples.

## 4. Conclusion

We proposed a framework to generate hard samples from the low-dimensional latent space of medical images. This framework showed improved data efficiency and can be applied to other tasks where a labeling tool is available (e.g. medical image segmentation).

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
