# OpenReview forum: "Suggestive Labelling for Medical Image Analysis by Adaptive Latent Space Sampling"
_MIDL.io/2020/Conference — Submitted to MIDL 2020_

### Official Review · AnonReviewer2 · 2020-03-12
**Suggestive Labeling for Medical Image Analysis by Adaptive Latent Space Sampling**

**Rating:** 1
**Confidence:** 5

**Review:**

Brief summary:
VAE is used to encode the raw image to some feature representation in a latent space, the annotation suggestion is done based on the latent space. The supervised training loss provides some gradients that reach the latent space. Such gradients are used for selecting the next batch of images for annotation.

Quality: Below average;
Clarity: Average;
Originality: New to me.
Significance: In terms of the experimental results, the improvement is not significant. The proposed method is compared to a simple random selection method. Supposedly, one should get much better results when comparing to random selection method.

Pros: interesting idea, interesting topic.
Cons: (1) Lack of comparisons with the state-of-the-art annotation suggestion methods. (2) The proposed method relies on gradient feedback from the supervised training loss for new sample selection. Such feedback only could give you some local movements in the latent space. In this sense, the selected samples might not be the most effective samples for the active learning task. (3) Sampled new point in the latent space may not correspond to a valid image sample after applying the decoder on it. Namely, the decoder can give you noisy "image samples".  (4) No strong justification that we should do annotation suggestion in the proposed way.

---

### Official Review · AnonReviewer3 · 2020-03-13
**Interesting idea; poor experimental results.**

**Rating:** 2
**Confidence:** 4

**Review:**

The paper presents an interesting idea which holds greater value in medical imaging as motivated in the paper. Not sure how much evaluation is expected in the short paper, but in my opinion, the paper lacks enough experimental evidence to support the key hypothesis. Moreover, as shown in Fig. 2b,  the improvement of the proposed method over baseline is not clear or not presented clearly.

Also, can authors present more examples of generated samples?

It could be understood that the current paper is more about the idea but the current model relies on the generative capacity of the model and VAE are well known for producing bad samples. The authors are suggested to consider [1] to improve sample quality or discuss the effect of sample quality in the current work.


[1] Diagnosing and Enhancing VAE Models [Dai and Wipf, 2019]

---

### Official Review · AnonReviewer4 · 2020-03-13

**Rating:** 2
**Confidence:** 4

**Review:**

This paper tackles an important question in machine learning on informative and efficient sampling of training data. This is somehow similar to a combination of active learning and hard negative mining, however, it is not justified in the paper what's the differences to the proposed method. In fact, the two areas are not mentioned at all.

pros:
+ the latent space sampling is interesting, which I believe is a promising direction, although not justified in the paper.

cons:
- the result shown in Figure 2(b) is not convincing to me, since both methods plateau quickly with only ~32 samples. The proposed approach has a very small 'effective window' which could diminish the impact of proposed approach.
- no comparison result provided to other hard negative mining methods.

---

### Official Review · AnonReviewer1 · 2020-03-13
**Review of Suggestive Labelling for Medical Image Analysis by Adaptive Latent Space Sampling**

**Rating:** 3
**Confidence:** 5

**Review:**


Summary:

The authors propose an active learning method using variational auto-encoder. They assess their method by predicting structural stress within vessel wall in Intravascular ultrasound.

Strengths:

-The idea of navigating in the latent space using the difference of the model’s predict with ground truth is original and makes sense.
-The article is well-written.

Weaknesses:

*I am unsure about the practical use of the method. The proposed method reaches its optimum performance at that same time than the baseline. You wouldn’t want to consciously use a suboptimal system for medical research, would you?
*The authors do not cite nor discuss relevant literature.
*Some details are unclear when they could have easily been added without using additional space.


Detailed comments:

“Experiment was repeated for five times for plotting the mean and variance (Figure 2(b))” I don’t see mean and variance in Figure 2.

“The neural network model F is first trained for a couple of epochs with the current training set using a given loss function. “ How many epochs?

“predefined size of training samples” What is the size?

The authors could cite more relevant literature instead of the computer vision datasets. For example:

Kingma, D.P. and Welling, M., 2013. Auto-encoding variational bayes. arXiv preprint arXiv:1312.6114.

Biffi, C., Oktay, O., Tarroni, G., Bai, W., De Marvao, A., Doumou, G., Rajchl, M., Bedair, R., Prasad, S., Cook, S. and O’Regan, D., 2018, September. Learning interpretable anatomical features through deep generative models: Application to cardiac remodeling. In International Conference on Medical Image Computing and Computer-Assisted Intervention (pp. 464-471). Springer, Cham.

---

### Meta-Review · Area_Chair1 · 2020-04-06
**MetaReview of Paper83 by AreaChair1**

**Rating:** 2

**Metareview:**

This short paper proposes an algorithm to select maximally useful batches for neural network training, based on the magnitude of the gradient as evaluated in a VAE latent space.

While the idea is interesting, and appreciated by the reviewers, the method does not appear to outperform random sampling, and the authors also do not compare to other label-efficient methods such as active learning.

**Paper Type:**

methodological development

---

### Decision · Program_Chairs · 2020-04-11

Reject